# Authoritative Practices and Collective Validation: Wikidata within the Collaborative Digital Edition of the Greek Anthology

Maxime Guénette       Mathilde Verstraete

Marcello Vitali-Rosati

**Keywords**: Greek Anthology, authority, collaboration, digital philology

The management and preservation of research data in the Humanities increasingly raises questions about its sustainability, sharing, and validation. In this context, Wikidata constitutes a powerful and collaborative tool. By challenging traditional models where researchers act as both producers and gatekeepers of authority, Wikidata redefines these issues and fosters new paradigms of collaboration.

This paper will explore these dynamics of collaboration and shifting authority through the case study of the collaborative digital edition of the *Greek Anthology* (the AG project, hosted at the Canada Research Chair on Digital Textualities since 2014), implemented on a collaborative platform (https://anthologiagraeca.org/) where everyone is invited to participate according to their own knowledge.

Wikidata is used in many ways within the AG project. First, all keywords (place names, authors, metrical forms, literary genres, etc.) used to annotate the platform have a Wikidata identifier or is created accordingly. Indeed, when a user participates in the editing of the corpus and wishes to add a keyword to an epigram, if the keyword does not exist, he or she must create it on Wikidata and then link it to the platform. Second, Wikidata has been used in a more intensive way to address inconsistencies in our list of

authors. Like Wikidata, our data model is multilingual. However, the gaps and inconsistencies in Wikidata —such as missing authors, duplicate entries, and inconsistent information across languages— were directly mirrored on our platform (https://anthologiagraeca.org/authors/). This alignment made it essential to tackle these issues systematically to ensure the accuracy of our data. We started by searching for the names of these authors in various languages (at least in French, English, Italian, Ancient Greek and Latin). We then uploaded this information to Wikidata, and subsequently fetched it back to integrate it into the AG platform. Almost immediately after our data dump on Wikidata, its community quickly reviewed and corrected it to align our contribution with Wikidata's standards and guidelines. This process means we not only retrieved our data but also benefited from the community's improvements.

We are making a conscious strategic choice: rather than positioning ourselves as the sole custodian of authority, we are delegating that responsibility to a wider community. Our presentation invites reflection on the implications of this shift toward distributed authority. How can that shift in authority benefit academic research projects? Is Wikidata's epistemological paradigm coherent with ours? Can we think of a generic epistemological framework to be effectively applied to specific academic endeavors?

Based on the experiments carried out and the choices made as part of the AG project, this presentation will provide practical and conceptual answers to the questions of (distributed) authority, validation and collaboration in the use of Wikidata, opening up prospects for other projects in the Humanities. We suggest that Wikidata is not merely a technical tool but rather a space where methodological and epistemological debates can unfold. By engaging with this dynamic, researchers can enhance their projects while contributing to the creation of a more sustainable, inclusive, and collaborative knowledge base.

