# OpenReview forum: "Authoritative Practices and Collective Validation: Wikidata within the Collaborative Digital Edition of the Greek Anthology"
_wikimedia.it/Wikidata_and_Research/2025/Conference — WD&R Paper_

### Official Review · ~Camillo_Carlo_Pellizzari_di_San_Girolamo1 · 2025-01-04
**A reflection on the use of Wikidata in academic research projects**

**Originality:** 5
**Impact:** 5
**Confidence:** 5

**Review:**

The authors describe how Wikidata is already being used as a source for multilingual authority data and tags by the Greek Anthology project (which is itself a collaborative project) and, starting from this experience, builds up more general reflections about the implications of delegating the curation of data used in academic research projects to a wider community.
These reflections can have a significant relevance in fostering the cooperation of other academic projects with Wikidata; this is particularly relevant in the field of digital humanities, where open access databases are still relatively rare but would be useful to open new research fields. One of the reasons that make it difficult to create and maintain such databases is the relevant amount of human and financial resources needed to curate them; the use of Wikidata, as shown here, should be considered among the ways to mitigate this problem.

**Compliance:**

5

**Scientific Quality:**

5

---

### Official Review · ~Monica_Berti1 · 2025-01-10
**An important contribution on the critical use of Wikidata for collaborative projects in the humanities and philology**

**Originality:** 5
**Impact:** 5
**Confidence:** 5

**Review:**

The authors of this paper aim not only to present the use of Wikidata in the collaborative digital edition of the Greek Anthology (the AG project), but also to discuss important questions about the relationship between Wikidata and research. These reflections are the result of the long experience of the authors, who have chosen to contribute data from their project to Wikidata, not only to preserve it in a wider, collaborative and distributed environment, but also to make it an opportunity for methodological and epistemological debates.

**Compliance:**

5

**Scientific Quality:**

5

---

### Decision · Program_Chairs · 2025-02-05

Accept (Paper)